

# Technical Note: Deciphering the Hydrologic Response of Riverbeds across Changes in Recharge with Electrical Resistivity Imaging

Weston J. Koehn [1], Stacey E. Tucker-Kulesza [2], and David R. Steward [3]

[1]Kansas State University 2118 Fiedler Hall Manhattan, KS 66506
[2]Kansas State University 2118 Fiedler Hall Manhattan, KS 66506
[3]Kansas State University 2118 Fiedler Hall Manhattan, KS 66506

*Correspondence to:* Weston Koehn (koehnweston@gmail.com)

**Abstract.** The fluxes between groundwater and surface water play a significant role in quantifying water balance along stream reaches to continent scales. Changes in these dynamics are occurring due to aquifer depletion, where river flow from predevelopment baseflow conditions with groundwater to surface water have evolved to enhanced recharge through streambeds of ephemeral flows to groundwater. This problem is studied along the Arkansas River in Western Kansas across a stream reach
that transitions from near equilibrium of fluxes to a losing river that contributes recharge to a depleting High Plains Aquifer. Existing hydrologic data illustrates the lack of understanding they provide related to the control of fluxes exerted by alluvial deposits. We employ electrical resistivity imaging (ERI) along this river transect to elucidate the intricate pathways of hydrologic connectivity existing between the Arkansas River and underlying Arkansas Alluvial and Ogallala Aquifers. Time-lapse ERI profiles quantify the temporal changes in resistivity across the riverbed, and these changes are associated with the distribution
of soil physical properties and hydrologic conditions below the water-sediment interface. Results utilize a recently discovered vadose zone property whereby fine grained inclusions may become revealed by their different water holding capacity relative to that of a surrounding matrix of coarser grained soil across changes in recharge (caused by changes in stream discharge). These findings corroborate the role of large-scale geologic features in maintaining streamflow in regions of near-surface impermeable layers, and the localized recharge that occurs to the High Plains Aquifer through embedded assemblages of fine and coarse
grained soils.

## 1 Introduction

Aquifer depletion contributes to an evolution in the hydrological exchanges between groundwater and surface water. This problem is studied in a region overlying the High Plains Aquifer where regional rivers, such as the Arkansas River in Western Kansas, were fed by groundwater prior to the development of widespread irrigated agriculture and the occurrence of depleting
groundwater stores (Gutentag et al., 1984). This groundwater system has crossed the threshold of peak groundwater depletion, where society is no longer capable of extracting the same levels of groundwater to sustain this agricultural region (Steward and Allen, 2016). Furthermore, the recharge occurring through the terrestrial farming ecosystem would require hundreds of years to replenish aquifer depletion by historical natural recharge processes (Steward et al., 2013). The losing rivers in this region





play an important role in the regional water balance as they serve as primary sources of groundwater recharge (Whittemore, 2002), and provide a source of recharge to the underlying Ogallala formation (Whittemore, 2002; Steward and Allen, 2016).

The recharge occurring beneath the ephemeral Arkansas River follows the flow regimes between surface water and groundwater typical of stream-aquifer interactions (Sophocleous, 2005; Brunner et al., 2009). The conceptual model in Figure 1,

illustrates four different connection regimes that may occur within a losing river environment, with a fully connected perennial river in figure 1a, and ephemeral conditions in the others. The groundwater/surface water system is fully coupled in figure 1b and a progression is illustrated where an unsaturated zone forms beneath the riverbed due to cessation of river flow in figure 1d. While the figure illustrates a gradually declining phreatic surface in a homogeneous vadose zone, a detailed understanding of the hydrogeologic properties within this region is needed to fully elucidate complex alluvial recharge processes (Sophocleous,

2002; Brunner et al., 2009).

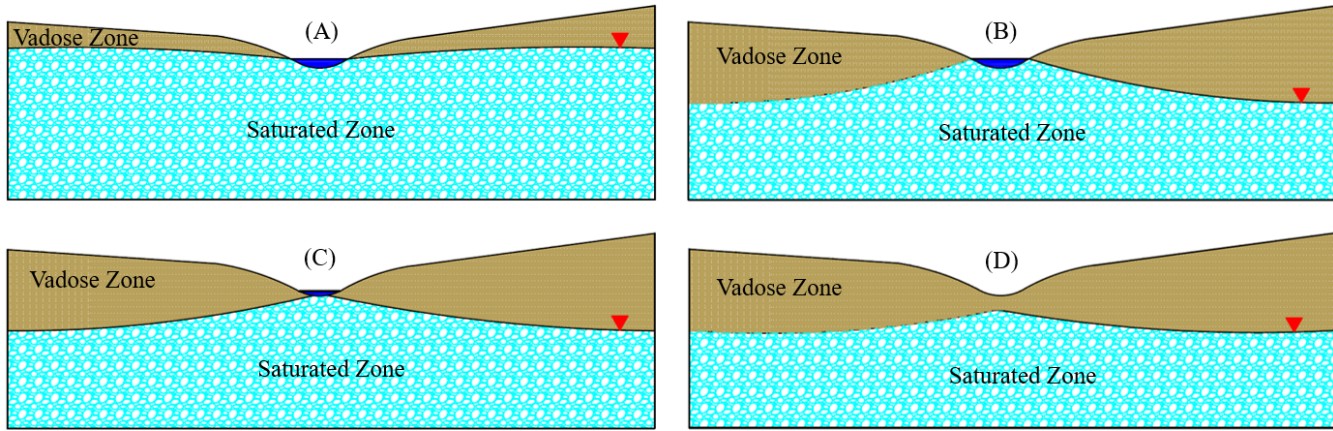

**Figure 1.** Schematic showing the different types of hydrologic connection between groundwater and surface water in baseflow and losing river environments;(A) stream is groundwater fed (B) streambed and saturated zone are fully connected (B) streambed and saturated zone may be partially connected, (C) streambed and saturated zone and are disconnected by a vadose zone.

This study examines the soil physical properties and hydrologic conditions directly below the water-sediment interface using a novel interpretation of electrical resistivity by incorporating results and insight gained from vadose zone modeling. Electrical resistivity is an intrinsic material property which quantifies how strongly a material can oppose the flow of electrical current. Hydrogeologic and environmental factors such as the water content, porosity, salinity, clay content, pore geometry,

and pore-fluid temperature control the electrical resistivity of a medium (Everett, 2013). ERI surveys can be used to delineate groundwater discharge areas (Nyquist et al., 2008), recharge pathways through mantled sinkholes (Schwartz and Schreiber, 2009), and riverbed sediment architecture (Crook et al., 2008). Daily et al. (1992) showed that ERI can effectively map changes in water content within the vadose zone by analyzing the temporal changes in electrical resistivity as a result of an infiltration event. Rucker (2009) coupled synthetic electrical resistivity models with an infiltration model to track the movement of a

wetting front through porous media. Ohara et al. (2018) conducted a series of ERI surveys to successfully delineate the spatial extent of an alluvial groundwater system within a mountainous region. Similarly, Watlet et al. (2018) evaluated the infiltration



dynamics within a karstic vadose zone with electrical resistivity tomography. These previous studies portray the efficacy of ERI to study vadose zone and groundwater hydrology, and support the application of such an approach to study alluvial recharge processes.

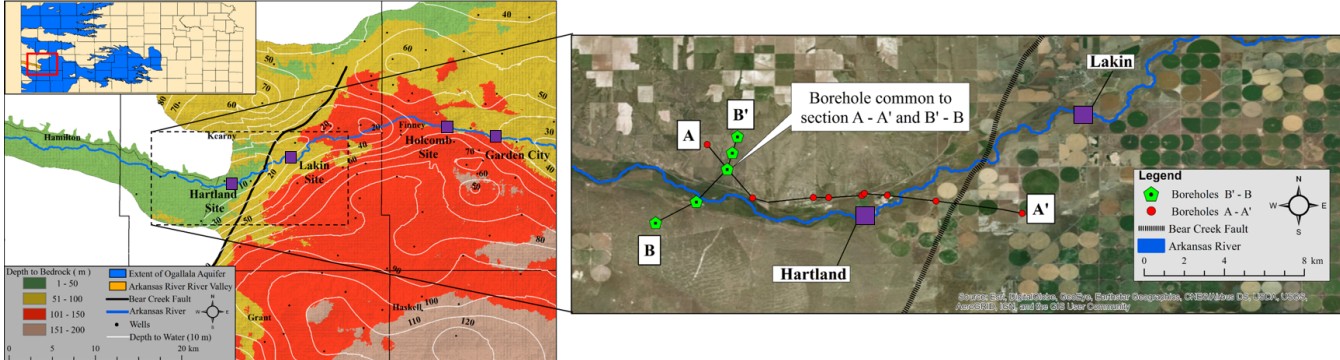

**Figure 2.** (A) Regional map of the study area showing the location of the Bear Creek fault in relation to the Arkansas River and the survey sites. The depth to bedrock and depth to water was interpolated from borehole records and monitoring well data provided by the Kansas Geological Survey

It is well known that the rate of recharge leaves a signature that is decipherable through measurements of pressure head in
the vadose zone (Pullan, 1990). Specifically, Steward (2016) showed that changes in recharge rate result in different patterns of pressure head distribution for layered soils with inclusions. A coarse soil embedded within a fine grained soil behaves differently than a fine grained soil embedded within a coarse grained soil, and these differences provide the perspective necessary to elucidate soil properties across changes in recharge rate (Steward, 2016). The existing well and borehole data across the region, shown shortly for this study, do not provide the level of detail necessary to define and detect riverbed heterogeneities
and hydrologic connection, both of which impact recharge processes and rates. This novel study maps soil properties with ERI across river corridor transects across changes in stream discharge to provide evidence of the geomorphic template that controls groundwater surface water interactions. Results are directly applicable to the advancement of physically based hydrologic models as they provide detailed insight into the subsurface hydrogeologic environment at the riverbed scale, and identify the type of hydrologic connection that exists below a losing river.

## 2 Hydrogeologic Setting and Methods

The Bear Creek fault, which is shown by the solid black line in Figure 1, is the key hydrogeologic feature within the study region. The Bear Creek fault is classified as a dissolution fault, as Johnson (1981) revealed that bedded salts of the Permian age were dissolved by infiltrating groundwater to depths as great as 250 m across regions of northwestern Oklahoma and southwestern Kansas. Seasonal bank storage provides baseflow to the Arkansas River along the river segment stretching between the
Kansas-Colorado State line and the Bear Creek fault (Whittemore, 2002). Analysis of streamflow data by Whittemore (2002) showed that substantial streamflow losses from the Arkansas River to groundwater occur between the river segment stretching



between Hartland and Garden City (downstream of the Bear Creek fault). The depth to water (white contours at 10 m intervals) and the depth to bedrock across the study region are given in Figure 2.

The heterogeneous alluvial sediments and large scale geologic features, such as the Bear Creek fault, are illustrated using geologic cross-sections constructed from existing borehole data in Figure 3. Existing lithological records from drilling logs

were analyzed, and their hydrogeologic units were consolidated into four categories. The cross section B' - B identifies a shallow alluvial system underlain by a thick shale and sandstone sequence upstream (west) of the Bear Creek fault, from north to south across the Arkansas River. The continuous shale and sandstone layer acts as the base of the Arkansas River Alluvial Aquifer west of the Bear Creek fault where the Ogallala formation is not present. The transition from a shallow alluvial system upstream (west) of the Bear Creek fault to a deeper multi-layered (Arkansas Alluvial and Ogallala) aquifer system downstream

(east) of the Bear Creek fault is illustrated and evident in cross section A - A', which is parallel to the Arkansas River and spans the Bear Creek fault from west to east.

Selection of three ERI survey sites along the Arkansas River were chosen to study the river connectivity of Figure 1 across different hydrogeologic conditions up-gradient and down-gradient of the Bear Creek Fault. Each site contained unique hydrologic and hydrogeologic aspects, such as the depth to bedrock, depth to water table, and connection between river and saturated

zone. The Hartland site overlies the Arkansas Alluvial Aquifer upstream of the Bear Creek fault where the Ogallala formation is not present. Bank storage provides baseflow to the river at Hartland during low flow periods (Whittemore, 2002). The Lakin site is located 6 km downstream of the Bear Creek fault, where substantial seepage losses from the river to underlying Arkansas Alluvial and Ogallala Aquifers are known to occur (Whittemore, 2007). The river and alluvial aquifer exhibit a transitional to full connection as the depth to water is generally less than 10 m within the alluvium and river discharge is fairly consistent.The

Holcomb site was chosen because of the highly periodic Arkansas River flows that occur there, which allowed for surveys to be conducted across a dry and inundated riverbed.

Electrical resistivity measurements are conducted by injecting electrical current into the subsurface through an electrode pair (current pair) while simultaneously measuring the induced voltage potential between a separate electrode pair (potential pair). Repeated sets of measurements using various electrode configurations are conducted in an ERI survey to collect an apparent

resistivity psuedosection. The SuperSting Earth Resistivity Induced Polarization and Self Potential System from Advanced Geosciences Inc. was used to conduct all ERI surveys in this study, and all data were processed with EarthImager 2D (AGI, 2007). Surveys were conducted parallel and perpendicular to the Arkansas River. Cross river surveys utilized submersible and dry electrode cables in series as the middle portions of those surveys were submerged. The temporal changes in resistivity were calculated using the inverted resistivity profiles.

## 30  3  Results and Discussion: Temporal Changes in Electrical Resistivity

Temporal changes in resistivity were calculated for survey transects located at Hartland, Lakin, and Holcomb, shown in Figure 2. These changes were analyzed across a reach of the Arkansas River that transitions from baseflow to seepage conditions. Changes in resistivity are illustrated for river transects of which are both fully connected and disconnected to the underlying





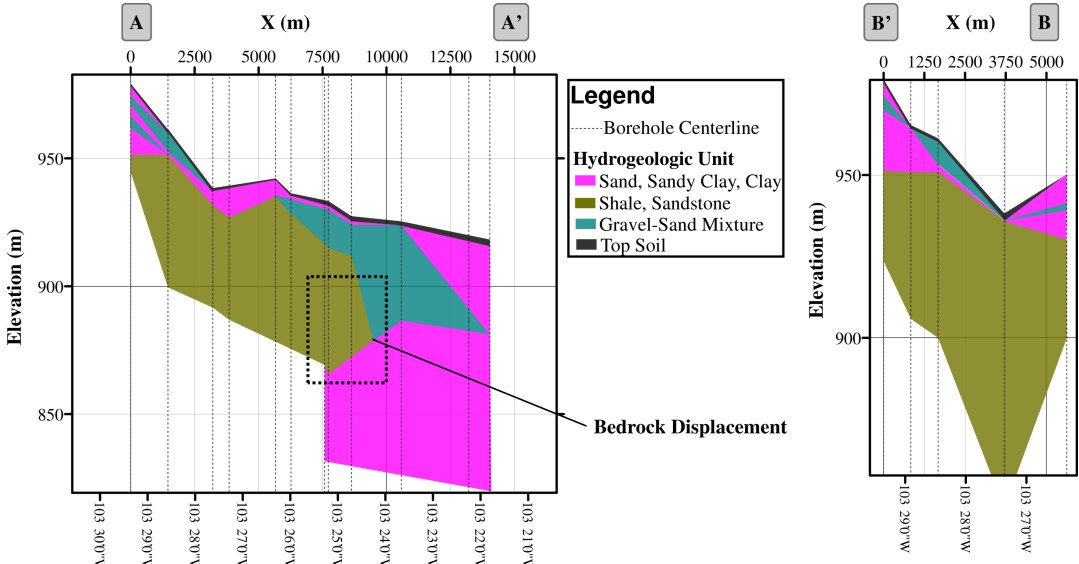

**Figure 3.** (Generalized geologic cross sections (A - A' and B' - B) created from existing borehole data aligned parallel and perpendicular to the Arkansas River and Bear Creek fault.

Arkansas Alluvial Aquifer (connection and disconnection defined by schematic in Figure 1). The ERI surveys at the Hartland site were conducted in September of 2015 and March of 2016, and the temporal changes in resistivity are shown in Figure 4C. The hydrologic conditions at Hartland follow those of Figure 1A, with a perennial stream and shallow groundwater table. A 10 $\Omega$ m to 20 $\Omega$ m increase in resistivity occurred between -1 m and -3 m, which is attributable to the depletion of bank

5 storage in response to lower river discharges throughout the winter months. Resistivity values decreased by 1 $\Omega$ m to 10 $\Omega$ m between -3 m and -7 m, which corresponds to the saturated portion of the Arkansas Alluvial Aquifer. These small changes are attributed changes in pore water resistivity. The Arkansas River is highly saline, and the resistivity of alluvial groundwater is highly dependent upon river discharge (Whittemore, 2000). The ERI does not reveal soil heterogeneity in the profile. Instead, the red zone (zone of increasing resistivity) is dewatered and the blue to purple zone (zone of decreasing resistivity) is more

10 saline with relatively homogeneous sediments.

   The next site at Lakin is down-gradient of the Bear Creek fault, and ERI surveys were conducted in July and September of 2016 with changes in resistivity shown in Figure 5. These surveys crossed the river, and the difference in resistivity between high and low flow periods (Figure 1b to Figure 1c) illustrate two recharge zones below the riverbed. The recharge zones experienced a 5 $\Omega$ m decrease in resistivity, and these changes within the saturated zone are in response to increases in pore

15 fluid salinity as a result of decreased river discharge. The Arkansas River generally exhibits higher salinity during low flow periods (Whittemore, 2000). The oblong shape and lateral spreading of the recharge zones (between 24 m and 84 m below -7 m) reveal anisotropic hydraulic properties and evidence of preferential flowpaths within the alluvium. This also indicates that a confining unit may exist between the alluvial aquifer and deeper underlying Ogallala aquifer, which would promote more





**Figure 4.** Hartland Site (Seasonal Baseflow Conditions); (A) ERI Survey in September 2015 ; (B) ERI Survey in March 2016; (C) Change in resistivity between September 2015 and March 2016. Note that the river is to the right, and no clear river crossings were available to conduct a cross-river survey.

lateral flow within the alluvial deposits. A dewatered zone is illustrated between 60 m and 84 m within near surface sediments above the groundwater table, and is attributable to evaporation and phreatophyte root water uptake.

The site at Holcomb is furthest down-gradient of the Bear Creek fault, and surveys were conducted during the midst of a high flow event (July 2016) and during no flow conditions (September 2016). This hydrologic regime crossed those of Figure 1b to 1d as the ephemeral river transitioned to a completely dry state between July and September. The changes in resistivity shown in Figure 6c illustrates dewatered sediments (blue to red zones) above a saturation front. Sediments containing highly saline pore fluid as a result of infiltrating river water are identified by the purple zones. The parabolic shaped inclusion between 20 m and 25 m reveals a region of fine grained soil with a different water holding capacity than the surrounding background.



**Figure 5.** Lakin Site (Transitional to Fully Connected Losing River) - (A) ERI Survey in July 2016; (B) ERI Survey in September 2016; (C) Change in resistivity between July 2016 and September 2016

Heterogeneity within the vadose zone has been shown to have a significant impact on vadose zone flow processes (Steward, 2016), and these results demonstrate the localized spatial scales that predominate control on alluvial recharge processes.

In this study, the spatial distribution soil physical properties and the existence of preferential flowpaths is revealed by analyzing the changes in electrical resistivity of riverbed sediments in response to changes river discharge. The desaturation of the vadose zone and infiltration of saline river water through permeable saturated sediments is clearly illustrated in the time difference profiles. The depth to the water the table, and connection between the water table and river is also illustrated through time by analyzing such changes. The methodology applied in this study provides a way to gain information on the hydrogeologic environment that exerts control on stream aquifer interactions across a localized scale.



**Figure 6.** Holcomb Site (Disconnected losing river); (A) ERI Survey in July 2016 ; (B) ERI Survey in September 2016; (C) Change in electrical resistivity between July 2016 and September 2016

## 4 Conclusions

Groundwater depletion in excess of natural recharge has resulted in the decline of groundwater levels within the Arkansas River Valley across Western Kansas over the past century. Consequently, the Arkansas River has transitioned from a groundwater fed stream to a losing stream. Recharge processes between the Arkansas River and the underlying Arkansas Alluvial Aquifer and Ogallala Aquifer are difficult to understand with the existing river gage, borehole, and well data. The dynamics of groundwater/surface water interactions are critical to water management studies, and missing information about the connection between the river and saturated zone are directly observed using the methodology applied here. The ERI results provide detailed delineation of preferential flow paths from surface water to groundwater across temporal dynamics in ephemeral and perennial river reaches. Within the vadose zone, positive changes in resistivity (increasing resistivity over time) were attributed



to the dewatering of localized regions within the vadose zone. Changes in resistivity below the groundwater table, attributable to temporal changes in pore water resistivity, illustrate the existence of preferential pathways within permeable and anisotropic alluvial sediments. Results illustrate how the hydrologic state within alluvial sediments responds to changes in river discharge, which significantly impacts the dynamic recharge process.

5    Fully integrated groundwater models are currently limited by the ability to conceptualize the hydrologic connection existing between surface water bodies and groundwater. Furthermore, the dynamic recharge process is driven by the evolution of the hydrologic state directly below the recharge source. ERI surveys of streambed sediments, and the observed hydrologic response to changes in river discharge provides the necessary insight to improve model conceptualization across different hydrologic regimes. Geophysical instrumentation of stream bottom sediments characterizes the controlling mechanisms of

10   river transmission loss to groundwater.

*Competing interests.*   No competing interests are present.

*Acknowledgements.*   Support for this project was provided by the USDA Ogallala Aquifer Project. We would also like to thank Southwest Kansas Groundwater Management District (GMD3) and the land owners for their assistance and cooperation.



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
