# Peer review of "Technical Note: Deciphering the Hydrologic Response of Riverbeds across Changes in Recharge with Electrical Resistivity Imaging"

_Hydrology and Earth System Sciences, 2018_

## Referee Comment (RC1) · Anonymous Referee #1 · 16 Apr 2018

General comment

The study presented in this manuscript describes time-lapse electrical resistivity imaging results associated with groundwater and surface water interactions in riverbeds, via data from three monitoring sites along the Arkansas River in Western Kansas. The research goals are well defined, i.e. investigating the different types of hydrologic connection between the groundwater and surface water via interpreting electrical resistivity changes as compared to vadose zone modeling. However significant methodological issues are not addressed and/or presented in details, which lead to critical concerns

about the time-lapse electrical resistivity imaging results. I could sum up these different concerns in the following list:

- No information on the types of electrodes used throughout the experiments is provided, especially given the unusual set up including river crossing.

- No information is given about the measurement protocols, and more specifically about the use of reciprocal measurements for assessing the measuring errors.

- No information is given on how the measuring errors have been assessed.

- There is also no information available about the parameters used for the inversions of the resistivity data, and especially on how the river water was taken into account in the inversion procedure.

- The topography is apparently not included in the inversion despite clearly visible slopes in the field pictures. Not including the topography could lead to artifacts in the resistivity image...

- There is also no explanations on the way temperature changes have been corrected despite the great impact they can induce on the resistivity of the subsurface, as it is well described for example in Brunet et al. (2010).

- Presenting the changes in resistivity (in Fig. 4, Fig 5. and Fig. 6) as absolute changes of resistivity without showing the background resistivity image is definitely not the best option as clearly explained in the review of Samouëlian et al. (2005). A variation of 10 Ohm.m within a 4000 Ohm.m area is not exactly similar to a variation of 10 Ohm.m within a 50 Ohm.m area... Moreover, given that no information is given about the error level of the measured voltages, which informs on the signal to noise ratio, this is hard to tell if such small variations of resistivity are actually interpretable...

Therefore, the interpretations of the changes in resistivity presented in the manuscript cannot really be trusted with a reasonable level of confidence. I also doubt that the chosen format (i.e. Technical Note) is relevant for presenting these results as this

manuscript does not present significant advances or novel experimental techniques. Imaging hydrological processes with time-lapse electrical resistivity imaging has already been addressed by several publications, including HESS papers, for the last 10 years. In summary, this paper has interesting goals and the electrical resistivity imaging techniques is an appropriate choice for addressing the associated scientific questions. However critical concerns are raised about the methodology applied for processing the data and visualizing the resistivity results. In my opinion, a significant amount of detailed information is still required for publishing this study. I would recommend a major revision of this manuscript, starting by better explaining the methodology used for producing the resistivity results. This will also most probably require from the authors an additional processing of the resistivity data to include at least the topography and corrections for temperature variations.

Specific comments

Introduction: The goals of the research are well presented, but some state of the art papers are missing. These include for example Binley et al. (2015), Chambers et al. (2014), Descloitres et al. (2008), Uhlemann et al. (2016), which could help the authors in exploring approaches for better presenting their results.

Figure 2: There is a (A) in the caption but no (B), while there is no (A) nor (B) in the figure. Including a view at the continental scale in the top left corner of the left subfigure could be more convenient. The font size of the legend in the bottom left corner is too small.

Figure 3: Linear interpolations between borehole logs are probably not the best ways to draw a geologic cross-section. Also, B and B' seem flipped compared to Figure 2 or the x axis has to be flipped in the right side of Figure 3.

Page 5: the interpreted changes in resistivity for the Hartland site or the Lakin site are generally very small: 10 to 20 Ohm.m (line 4), 1 Ohm.m to 10 Ohm.m (line 5), 5 Ohm.m (line 14). These can be attributed either to artifacts from the inversion, noise

in the measured voltage or temperature variations instead of actual changes in soil moisture content. Presenting changes in resistivity as resistivity ratios as it is generally the case in other studies would be much more convenient to evaluate whether this changes mean anything related to hydrological processes.

Figure 5: the changes in resistivity interpreted as recharge zones are so small that they could be associated to anything else than actual recharge... No topography is included in the resistivity model while slopes are clearly visible at the river banks in pictures in (A) and (B).

Page 5 line 8: "The ERI does not reveal soil heterogeneity in the profile". I would like to trust you but it would be easier if the resistivity of each site was shown in the figures.

Page 6 & 7: The changes in resistivity measured at the Hollocomb site are larger than for the other sites which make them more reliable, even if the lack of information concerning how the temperature change was taken into account is still problematic for initiating further interpretations. Discussing why these changes in resistivity are larger than for other sites could also be helpful to understand the different hydrological processes that the paper aims to describe.

Technical corrections

Page 5 line 7: attributed to

Caption of Figure 4: "Note that the river is to the right". To the right of the pictures or the resistivity model?

Page 7 line 6: the depth to the water table

References: Binley, A., Hubbard, S. S., Huisman, J. A., Revil, A., Robinson, D. A., Singha, K. and Slater, L. D.: The emergence of hydrogeophysics for improved understanding of subsurface processes over multiple scales: The Emergence of Hydrogeophysics, Water Resources Research, 51(6), 3837–3866, doi:10.1002/2015WR017016, 2015.

Chambers, J. E., Wilkinson, P. B., Uhlemann, S., Sorensen, J. P. R., Roberts, C., Newell, A. J., Ward, W. O. C., Binley, A., Williams, P. J., Gooddy, D. C., Old, G. and Bai, L.: Derivation of lowland riparian wetland deposit architecture using geophysical image analysis and interface detection, Water Resources Research, 50(7), 5886–5905, doi:10.1002/2014WR015643, 2014.

Descloitres, M., Ruiz, L., Sekhar, M., Legchenko, A., Braun, J.-J., Mohan Kumar, M. S. and Subramanian, S.: Characterization of seasonal local recharge using electrical resistivity tomography and magnetic resonance sounding, Hydrological Processes, 22(3), 384–394, doi:10.1002/hyp.6608, 2008.

Brunet, P., Clément, R. and Bouvier, C.: Monitoring soil water content and deficit using Electrical Resistivity Tomography (ERT) – A case study in the Cevennes area, France, Journal of Hydrology, 380(1–2), 146–153, doi:10.1016/j.jhydrol.2009.10.032, 2010.

Samouëlian, A., Cousin, I., Tabbagh, A., Bruand, A. and Richard, G.: Electrical resistivity survey in soil science: a review, Soil and Tillage Research, 83(2), 173–193, doi:10.1016/j.still.2004.10.004, 2005.

Uhlemann, S. S., Sorensen, J. P. R., House, A. R., Wilkinson, P. B., Roberts, C., Gooddy, D. C., Binley, A. M. and Chambers, J. E.: Integrated time-lapse geoelectrical imaging of wetland hydrological processes, Water Resources Research, 52(3), 1607–1625, doi:10.1002/2015WR017932, 2016.

---

## Referee Comment (RC2) · Anonymous Referee #2 · 25 Apr 2018

The presented manuscript describe the use of Electrical Resistivity Imaging (ERI) for imaging seasonal resistivity changes for three profiles at the Arkansas river. Aim is to get insight into recharge and discharge mechanisms, a very interesting question for which ERI is an appropriate method to be used along with others to gain understanding. After describing the hydrological setting, they spend a few sentences on the ERI method, the used instruments and cables. In the result section they show for each profile some pictures along with an image of the absolute resistivity changed inferred from the inversion of the measurements. The conclusion consists of four already-known

statements, four very general ERI interpretations and four outlooking sentences.

I am not familiar with the rare manuscript type "Technical Note" and how the evaluation differs from a normal paper. The instructions describe it as "Technical notes report new developments, significant advances, and novel aspects of experimental and theoretical methods and techniques"

I understand that for a Note a paper can (or must) be short but would expect some novel approach in either the measurement type, the data analysis or the understanding of processes. Neither of this seems to be the case. Additionally, more technical details are required, including the electrode spread, the used measuring protocol, and the inversion options (or approach) in order to understand the relevance of the results. The authors are apparently no experts in ERI. Apart from the vast literature in Applied Geophysics, there is a large number of ERI (particularly time-lapse) papers in hydrology journals (including HESS) that are widely ignored, some being very similar relevant. Specifically, there have been papers on monitoring river water - ground water interactions giving detailed background and conclusions. I see no lessons to be learned (significance), maybe only for the very specific site. But I see a lot of shortcomings in the description of the methodology that make it impossible to assess scientific quality. The organization is appropriate and the texts are well-written.

To summarize, I cannot recommend publication of the manuscript in the high-standard journal HESS, mainly due to lack of novelty and significance. In case a resubmission is recommended, a (very major) revision should include i) necessary details, ii) absolute resistivity tomograms as well as relative differences, iii) a thorough literature review and iv) a critical discussion of the results that go beyond the findings specific for your study area. In this case I would offer my services as reviewer.

Other comments

A Profile layout is missing, you just said that one profile is parallel and one is perpendicular. At least for the parallel it is questionable whether 2D conditions (constant

conductivity perpendicular to the inversion plane, i.e. topography) are met and if not how this could affect your results.

Details on the data analysis and results (data fit etc.) missing.

How did the river water conductivity change over time? This also includes temperature changes. Why did you not account for it in the interpretation or conclusions?

How was the river body treated in the inversion routine?

Particularly for the Lakin Site, the geometry did change as both the water level and the width of the river changed. How did you compare

Figures 4+5: You show absolute resistivity changes, which is is not really meaningful and therefore not used in literature. Please use relative change (in percent or as a ratio) that can be transferred directly into relative saturation (Brunet et al.).

Figure 1: denotation of subfigures is wrong: (B) streambed and saturated zone may be partially connected ==> (C) (C) streambed and saturated zone and are disconnected by a vadose zone => (D) Text uses Figure 1a etc., please be consistent.

Figure 5: very slight decreases are interpreted as huge recharge zones. However, I doubt that the typical resolution measures allow for such an interpretation. At least a critical discussion is missing.

Except the very recent paper of Watlet, papers on ERI in hydrology are extremely rare and rather old (>10 years). In the recent years there have been a number of papers, particularly in the HESS journal:

- Quantifying shallow subsurface water and heat dynamics using coupled hydrological-thermal-geophysical inversion, Anh Phuong Tran, Baptiste Dafflon, Susan S. Hubbard, Michael B. Kowalsky, Philip Long, Tetsu K. Tokunaga, and Kenneth H. Williams, Hydrol. Earth Syst. Sci., 20, 3477-3491, https://doi.org/10.5194/hess-20-3477-2016, 2016 - Monitoring hillslope moisture dynamics with surface ERT for enhancing spatial signif-

icance of hydrometric point measurements, R. Hübner, K. Heller, T. Günther, and A. Kleber, Hydrol. Earth Syst. Sci., 19, 225-240, https://doi.org/10.5194/hess-19-225-2015, 2015 - Three-dimensional monitoring of soil water content in a maize field using Electrical Resistivity Tomography L. Beff, T. Günther, B. Vandoorne, V. Couvreur, and M. Javaux, Hydrol. Earth Syst. Sci., 17, 595-609, https://doi.org/10.5194/hess-17-595-2013, 2013 - A geophysical analysis of hydro-geomorphic controls within a headwater wetland in a granitic landscape, through ERI and IP, E. S. Riddell, S. A. Lorentz, and D. C. Kotze, Hydrol. Earth Syst. Sci., 14, 1697-1713, https://doi.org/10.5194/hess-14-1697-2010, 2010

Other hydrology journals - Robinson, D. A., A. Binley, N. Crook, F. D. Day-Lewis, T. P. A. Ferre ', V. J. S. Grauch, R. Knight, M. Knoll, V. Lakshmi, R. Miller, J. - Nyquist, L. Pellerin, K. Singha, and L. Slater, 2008, Advancing process-based watershed hydrological research using near-surface geophysics: A vision for, and review of, electrical and magnetic geophysical methods: Hydrological Processes, 22, 3604–3635, doi: 10.1002/hyp.6963. - Ward, A. S., M. N. Gooseff, and K. Singha, 2010, Imaging hyporheic zone solute transport using electrical resistivity: Hydrological Processes, 24, 948–953, doi: 10.1002/hyp.7672.

Outside of Hydrology: - Coscia, I., Greenhalgh, S., Linde, N., Doetsch, J. A., Marescot, L., Günther, T., Vogt, T. & Green, A. (2011): 3D crosshole ERT for aquifer characterization and monitoring of infiltrating river water. Geophysics 76(2), G49-G59, doi:10.1190/1.3553003.

There are also a couple of papers in HESS on groundwater/river water interaction without ERI being involved.

---

## Referee Comment (RC3) · Anonymous Referee #3 · 22 May 2018

**Technical Note: Deciphering the Hydrologic Response of Riverbeds across Changes in Recharge with Electrical Resistivity Imaging**

Weston J. Koehn1, Stacey E. Tucker-Kulesza2, and David R. Steward3

1Kansas State University 2118 Fiedler Hall Manhattan, KS 66506

2Kansas State University 2118 Fiedler Hall Manhattan, KS 66506

3Kansas State University 2118 Fiedler Hall Manhattan, KS 66506

*Correspondence to:* Weston Koehn (koehnweston@gmail.com)

Abstract. The fluxes between groundwater and surface water play a significant role in quantifying water balance along stream reaches to continent scales. Changes in these dynamics are occurring due to aquifer depletion, where river flow from predevelopment baseflow conditions with groundwater to surface water have evolved to enhanced recharge through streambeds of ephemeral flows to groundwater. This problem is studied along the Arkansas River in Western Kansas across a stream reach

- 5 that transitions from near equilibrium of fluxes to a losing river that contributes recharge to a depleting High Plains Aquifer. Existing hydrologic data illustrates the lack of understanding they provide related to the control of fluxes exerted by alluvial deposits. We employ electrical resistivity imaging (ERI) along this river transect to elucidate the intricate pathways of hydrologic connectivity existing between the Arkansas River and underlying Arkansas Alluvial and Ogallala Aquifers. Time-lapse ERI profiles quantify the temporal changes in resistivity across the riverbed, and these changes are associated with the distribution
- 10 of soil physical properties and hydrologic conditions below the water-sediment interface. Results utilize a recently discovered vadose zone property whereby fine grained inclusions may become revealed by their different water holding capacity relative to that of a surrounding matrix of coarser grained soil across changes in recharge (caused by changes in stream discharge). These findings corroborate the role of large-scale geologic features in maintaining streamflow in regions of near-surface impermeable layers, and the localized recharge that occurs to the High Plains Aquifer through embedded assemblages of fine and coarse
- 15 grained soils.

**1 Introduction**

Aquifer depletion contributes to an evolution in the hydrological exchanges between groundwater and surface water. This problem is studied in a region overlying the High Plains Aquifer where regional rivers, such as the Arkansas River in Western Kansas, were fed by groundwater prior to the development of widespread irrigated agriculture and the occurrence of depleting

20 groundwater stores (Gutentag et al., 1984). This groundwater system has crossed the threshold of peak groundwater depletion, where society is no longer capable of extracting the same levels of groundwater to sustain this agricultural region (Steward and Allen, 2016). Furthermore, the recharge occurring through the terrestrial farming ecosystem would require hundreds of years to replenish aquifer depletion by historical natural recharge processes (Steward et al., 2013). The losing rivers in this region

play an important role in the regional water balance as they serve as primary sources of groundwater recharge (Whittemore, 2002), and provide a source of recharge to the underlying Ogallala formation (Whittemore, 2002; Steward and Allen, 2016).

The recharge occurring beneath the ephemeral Arkansas River follows the flow regimes between surface water and groundwater typical of stream-aquifer interactions (Sophocleous, 2005; Brunner et al., 2009). The conceptual model in Figure 1,

5 illustrates four different connection regimes that may occur within a losing river environment, with a fully connected perennial river in figure 1a, and ephemeral conditions in the others. The groundwater/surface water system is fully coupled in figure 1b and a progression is illustrated where an unsaturated zone forms beneath the riverbed due to cessation of river flow in figure 1d. While the figure illustrates a gradually declining phreatic surface in a homogeneous vadose zone, a detailed understanding of the hydrogeologic properties within this region is needed to fully elucidate complex alluvial recharge processes (Sophocleous, 2002; Brunner et al., 2009).

---

## Author Comment (AC1) · 19 Jun 2018

**General comments:**

The study presented in this manuscript describes time-lapse electrical resistivity imaging results associated with groundwater and surface water interactions in riverbeds, via data from three monitoring sites along the Arkansas River in Western Kansas. The research goals are well defined, i.e. investigating the different types of hydrologic connection between the groundwater and surface water via interpreting electrical resistivity changes as compared to vadose zone modeling. However significant methodological issues are not addressed and/or presented in details, which lead to critical concerns about the time-lapse electrical resistivity imaging results. I could sum up these different concerns in the following list:

**Author Response:** Thank you for the thorough review and mindful comments. Major revisions will include the following:

Expanded methodology to include: ERI survey setup, inversion criteria, model quality, and details about the time-lapse profiles

Added discussion to better explain how the ERI surveys can be used to develop new understanding of complex river-aquifer interactions both locally and regionally.

**RC1:** No information on the types of electrodes used throughout the experiments is provided, especially given the unusual set up including river crossing.

**Author Response:** A submersible electrode cable was used for portions of surveys crossing the river. Dry electrode cables were used for land based survey segments.

**RC2:** No information is given about the measurement protocols, and more specifically about the use of reciprocal measurements for assessing the measuring errors.

**Author Response:** . Reciprocal measurements were conducted to ensure that quality data were used for the inversions. Additional methodological details will be added to the revised manuscript.

**RC3:** No information is given on how the measuring errors have been assessed.

**Author Response:** . Survey details will be added to the revised manuscript.

**RC4:** There is also no information available about the parameters used for the inversions of the resistivity data, and especially on how the river water was taken into account in the inversion procedure.

**Author Response:** . The river water resistivity was measured at the time of each underwater survey, and used as an inversion parameter for underwater surveys. Specific details about the inversion and handling of the river water resistivity will be added to the methodology section within the revised manuscript.

**RC5:** The topography is apparently not included in the inversion despite clearly visible slopes in the field pictures. Not including the topography could lead to artifacts in the resistivity image. . .

**Author Response:** . The topography was included within the inverted resistivity sections however, the topography was not included within the time-lapse profile due to software limitations. The topography will be included within the images in the revised manuscript.

**RC6:** There is also no explanations on the way temperature changes have been corrected despite the great impact they can induce on the resistivity of the subsurface, as it is well described for example in Brunet et al. (2010).

**Author Response:** . No corrections for temperature changes were made for regions below the ground surface. However, the pore fluid resistivity was measured at the time of each survey, and used as an inversion parameter for submerged survey segments.

**RC7:** Presenting the changes in resistivity (in Fig. 4, Fig 5. and Fig. 6) as absolute changes of resistivity without showing the background resistivity image is definitely not the best option as clearly explained in the review of Samouëlian et al. (2005). A variation of 10 Ohm.m within a 4000 Ohm.m area is not exactly similar to a variation of 10 Ohm.m within a 50 Ohm.m area. . . Moreover, given that no information is given about the error level of the measured voltages, which informs on the signal to noise ratio, this is hard to tell if such small variations of resistivity are actually interpretable. . . Therefore, the interpretations of the changes in resistivity presented in the manuscript cannot really be trusted with a reasonable level of confidence. I also doubt that the chosen format (i.e. Technical Note) is relevant for presenting these results as this manuscript does not present significant advances or novel experimental techniques. Imaging hydrological processes with time-lapse electrical resistivity imaging has already been addressed by several publications, including HESS papers, for the last 10 years. In summary, this paper has interesting goals and the electrical resistivity imaging techniques is an appropriate choice for addressing the associated scientific questions. However critical concerns are raised about the methodology applied for processing the data and visualizing the resistivity results. In my opinion, a significant amount of detailed information is still required for publishing this study. I would recommend a major revision of this manuscript, starting by better explaining the methodology used for producing the resistivity results. This will also most probably require from the authors an additional processing of the resistivity data to include at least the topography and corrections for temperature variations.

**Author Response:** We revised a full manuscript w/ details to a technical note at recommendation of editor. This removed lots of geophysical details (including background resistivity images). We will include as much detail as possible while staying within the technical note page limit. A major revision of the methodology section will be conducted to better explain the setup and processing of the geophysical results. The revised manuscript will present the changes in resistivity as ratios.

Details about the inversion quality and measurement errors will also be added. A detailed discussion regarding temperature variations will also be added.

**Specific comments:**

**RC9:** Introduction: The goals of the research are well presented, but some state of the art papers are missing. These include for example Binley et al. (2015), Chambers et al. (2014), Descloitres et al. (2008), Uhlemann et al. (2016), which could help the authors in exploring approaches for better presenting their results.

**Author Response:** The introduction will be rewritten to include the novel aspects within the suggested references.

**RC10:** Figure 2: There is a (A) in the caption but no (B), while there is no (A) nor (B) in the figure. Including a view at the continental scale in the top left corner of the left subfigure could be more convenient. The font size of the legend in the bottom left corner is too small.

**Author Response:** Figure 2 will be reformatted as suggested.

**RC11:** Figure 3: Linear interpolations between borehole logs are probably not the best ways to draw a geologic cross-section. Also, B and B' seem flipped compared to Figure 2 or the x axis has to be flipped in the right side of Figure 3.

**Author Response:** The borehole to borehole cross section were originally constructed using eight different geologic units. The use of geospatial interpolation) techniques (Inverse Distance Weighting and Kriging) to create soil horizons yielded unrealistic cross sections due to the high level of geologic heterogeneity. Therefore, all geologic units were grouped into the categories shown in Figure 3 to allow for the creation of generalized cross sections by manual linear interpolation.

**RC12:** Page 5: the interpreted changes in resistivity for the Hartland site or the Lakin site are generally very small: 10 to 20 Ohm.m (line 4), 1 Ohm.m to 10 Ohm.m (line 5), 5 Ohm.m (line 14). These can be attributed either to artifacts from the inversion, noise in the measured voltage or temperature variations instead of actual changes in soil moisture content. Presenting changes in resistivity as resistivity ratios as it is generally the case in other studies would be much more convenient to evaluate whether this changes mean anything related to hydrological processes.

**Author Response:** The time-lapse profiles will be reprocessed and presented as resistivity ratios to improve interpretability. A detailed discussion will be added to explain the merit of the geophysical interpretations as they relate to hydrologic processes.

**RC13:** Figure 5: the changes in resistivity interpreted as recharge zones are so small that they could be associated to anything else than actual recharge. . . No topography is included in the resistivity model while slopes are clearly visible at the riverbanks in pictures in

(A) and (B).
(B)

**Author Response:** The time-lapse images will be reprocessed to show the topography that was used to invert the background resistivity profiles.

**RC14:** Page 5 line 8: "The ERI does not reveal soil heterogeneity in the profile". I would like to trust you but it would be easier if the resistivity of each site was shown in the figures.

**Author Response:** The background resistivity images will not be shown within the revised manuscript due to page limits however, the topography and inversion quality will be included to improve quality and interpretability.

**RC15:** Page 6 & 7: The changes in resistivity measured at the Holcomb site are larger than for the other sites, which make them more reliable, even if the lack of information concerning how the temperature change was taken into account is still problematic for initiating further interpretations. Discussing why these changes in resistivity are larger than for other sites could also be helpful to understand the different hydrological processes that the paper aims to describe.

**Author Response:** Additional discussion about the temporal changes in resistivity will be added to better emphasize their role in understanding hydrologic processes.

**Technical corrections**

**RC16:** Page 5 line 7: attributed to

**Author Response:** The correction will be made within the revised manuscript.

**RC17:** Caption of Figure 4: "Note that the river is to the right". To the right of the pictures or the resistivity model?

**Author Response:** The corrections will be made within the revised manuscript.

**RC18:** Page 7 line 6: the depth to the water table

**Author Response:** The correction will be made within the manuscript.

**Suggested References:**

Binley, A., Hubbard, S. S., Huisman, J. A., Revil, A., Robinson, D. A., Singha, K. and Slater, L. D.: The emergence of hydrogeophysics for improved understanding of subsurface processes over multiple scales: The Emergence of Hydrogeophysics, Water Resources Research, 51(6), 3837–3866, doi:10.1002/2015WR017016, 2015.

Chambers, J. E., Wilkinson, P. B., Uhlemann, S., Sorensen, J. P. R., Roberts, C., Newell, A. J., Ward, W. O. C., Binley, A., Williams, P. J., Gooddy, D. C., Old, G. and Bai, L.: Derivation of lowland riparian wetland deposit architecture using geophysical image analysis and interface detection, Water Resources Research, 50(7), 5886–5905, doi:10.1002/2014WR015643, 2014.

Descloitres, M., Ruiz, L., Sekhar, M., Legchenko, A., Braun, J.-J., Mohan Kumar, M. S. and Subramanian, S.: Characterization of seasonal local recharge using electrical resistivity tomography and magnetic resonance sounding, Hydrological Processes, 22(3), 384–394, doi:10.1002/hyp.6608, 2008.

Brunet, P., Clément, R. and Bouvier, C.: Monitoring soil water content and deficit using Electrical Resistivity Tomography (ERT) – A case study in the Cevennes area, France, Journal of Hydrology, 380(1–2), 146–153, doi:10.1016/j.jhydrol.2009.10.032, 2010.

Samouëlian, A., Cousin, I., Tabbagh, A., Bruand, A. and Richard, G.: Electrical resistivity survey in soil science: a review, Soil and Tillage Research, 83(2), 173–193, doi:10.1016/j.still.2004.10.004, 2005.

Uhlemann, S. S., Sorensen, J. P. R., House, A. R., Wilkinson, P. B., Roberts, C., Gooddy, D. C., Binley, A. M. and Chambers, J. E.: Integrated time-lapse geoelectrical imaging of wetland hydrological processes, Water Resources Research, 52(3), 1607–1625, doi:10.1002/2015WR017932, 2016.

---

## Author Comment (AC2) · 19 Jun 2018

**General comments:**

The presented manuscript describe the use of Electrical Resistivity Imaging (ERI) for imaging seasonal resistivity changes for three profiles at the Arkansas river. Aim is to get insight into recharge and discharge mechanisms, a very interesting question for which ERI is an appropriate method to be used along with others to gain understanding. After describing the hydrological setting, they spend a few sentences on the ERI method, the used instruments and cables. In the result section they show for each profile some pictures along with an image of the absolute resistivity changed inferred from the inversion of the measurements. The conclusion consists of four already-known statements, four very general ERI interpretations and four outlooking sentences. I am not familiar with the rare manuscript type "Technical Note" and how the evaluation differs from a normal paper. The instructions describe it as "Technical notes report new developments, significant advances, and novel aspects of experimental and theoretical methods and techniques" I understand that for a Note a paper can (or must) be short but would expect some novel approach in either the measurement type, the data analysis or the understanding of processes. Neither of this seems to be the case. Additionally, more technical details are required, including the electrode spread, the used measuring protocol, and the inversion options (or approach) in order to understand the relevance of the results. The authors are apparently no experts in ERI. Apart from the vast literature in Applied Geophysics, there is a large number of ERI (particularly time-lapse) papers in hydrology journals (including HESS) that are widely ignored, some being very similar relevant. Specifically, there have been papers on monitoring river water – ground water interactions giving detailed background and conclusions. I see no lessons to be learned (significance), maybe only for the very specific site. But I see a lot of shortcomings in the description of the methodology that make it impossible to assess scientific quality. The organization is appropriate and the texts are well-written. To summarize, I cannot recommend publication of the manuscript in the high-standard journal HESS, mainly due to lack of novelty and significance. In case a resubmission is recommended, a (very major) revision should include i) necessary details, ii) absolute resistivity tomograms as well as relative differences, iii) a thorough literature review and iv) a critical discussion of the results that go beyond the findings specific for your study area. In this case I would offer my services as reviewer.

**Author Response:**

We thank you for your thorough review and constructive comments. Major revisions will be made to the manuscript, which will be resubmitted as a new submission. Specifically, a detailed discussion about the understanding developed from this research as it pertains to river-aquifer interactions will be added to the revised manuscript to better highlight the hydrologic importance of this work. Additionally, significant details about the geophysical instrumentation and inversion criteria will be added to the methodology section to provide evidence of scientific quality. An extensive review of studies addressing groundwater-surface water interactions will

be conducted to further examine the previous work conducted within this area, and to shed light on the novel aspects of this work.

**Specific Comments:**

**RC1:** A Profile layout is missing, you just said that one profile is parallel and one is perpendicular.

**Author Response:** A figure showing the location of the ERI profiles in relation to the Arkansas River will be added to the revised manuscript.

**RC2:** At least for the parallel it is questionable whether 2D conditions (constant conductivity perpendicular to the inversion plane, i.e. topography) are met and if not how this could affect your results.

**Author Response:** The initial goal was to conduct surveys both parallel and perpendicular to the river at the Hartland site however, no survey could be conducted perpendicular to the river due to heavy vegetation.

**RC3:** Details on the data analysis and results (data fit etc.) missing.

**Author Response:** A section regarding details about the inversion criteria will be added to the revised manuscript.

**RC4:** How did the river water conductivity change over time? This also includes temperature changes. Why did you not account for it in the interpretation or conclusions?

**Author Response:** The river water resistivity was measured at the time of each survey as used as an input parameter in the inversions. The revised manuscript will report the river water resistivity at the time of each survey.

**RC5:** How was the river body treated in the inversion routine?

**Author Response:** The river body was treated as a constant resistivity layer (measured resistivity of river water in the field using conductivity probe). The depth of the river was surveyed at each electrode. A section regarding the underwater inversion criteria will be added to the methodology section of the revised manuscript.

**RC6:** Particularly for the Lakin Site, the geometry did change as both the water level and the width of the river changed. How did you compare Figures 4+5: You show absolute resistivity changes, which is is not really meaningful and therefore not used in literature. Please use relative change (in percent or as a ratio) that can be transferred directly into relative saturation (Brunet et al.).

**Author Response:** The revised manuscript will present all time-lapse profiles using a ratio of relative change. Because we do not know the spatial continuity of pore-fluid resistivity, it is not possible to directly infer relative saturation from these ERI surveys.

**RC7:** Figure 1: denotation of subfigures is wrong: (B) streambed and saturated zone may be partially connected ==> (C) (C) streambed and saturated zone and are disconnected by a vadose zone => (D) Text uses Figure 1a etc., please be consistent.

**Author Response:** The changes will be made within the revised manuscript.

**RC8:** Figure 5: very slight decreases are interpreted as huge recharge zones. However, I doubt that the typical resolution measures allow for such an interpretation. At least a critical discussion is missing.

**Author Response:** A detailed discussion about the interpretation of changes in resistivity as they relate to hydrologic processes will be added to the revised manuscript.

**RC9:** Except the very recent paper of Watlet, papers on ERI in hydrology are extremely rare and rather old (>10 years). In the recent years there have been a number of papers, particularly in the HESS journal:

**Author Response:** An extensive review of studies addressing groundwater-surface water interactions (as will the suggested references) will be conducted to further examine the impact of previous work conducted within this area, and to shed light on the novel aspects of this work.

**Suggested References:**

- Quantifying shallow subsurface water and heat dynamics using coupled hydrologicalthermal-geophysical inversion, Anh Phuong Tran, Baptiste Dafflon, Susan S. Hubbard, Michael B. Kowalsky, Philip Long, Tetsu K. Tokunaga, and Kenneth H. Williams, Hydrol. Earth Syst. Sci., 20, 3477-3491, https://doi.org/10.5194/hess-20-3477-2016, 2016

Monitoring hillslope moisture dynamics with surface ERT for enhancing spatial signifC3 icance of hydrometric point measurements, R. Hübner, K. Heller, T. Günther, and A. Kleber, Hydrol. Earth Syst. Sci., 19, 225-240, https://doi.org/10.5194/hess-19-225-2015, 2015

Three-dimensional monitoring of soil water content in a maize field using Electrical Resistivity Tomography L. Beff, T. Günther, B. Vandoorne, V. Couvreur, and M. Javaux, Hydrol. Earth Syst. Sci., 17, 595-609, https://doi.org/10.5194/hess-17-595-2013, 2013

A geophysical analysis of hydro-geomorphic controls within a headwater wetland in a granitic landscape, through ERI and IP, E. S. Riddell, S. A. Lorentz, and D. C. Kotze, Hydrol. Earth Syst. Sci., 14, 1697-1713, https://doi.org/10.5194/hess-14- 1697-2010, 2010

**Other hydrology journals –**

Robinson, D. A., A. Binley, N. Crook, F. D. Day-Lewis, T. P. A. Ferre 0, V. J. S. Grauch, R. Knight, M. Knoll, V. Lakshmi, R. Miller, J. - Nyquist, L. Pellerin, K. Singha, and L. Slater, 2008, Advancing process-based watershed hydrological research using near-surface geophysics: A vision for, and review of, electrical and magnetic geophysical methods: Hydrological Processes, 22, 3604–3635, doi:10.1002/hyp.6963.

Ward, A. S., M. N. Gooseff, and K. Singha, 2010, Imaging hyporheic zone solute transport using electrical resistivity: Hydrological Processes, 24, 948–953, doi: 10.1002/hyp.7672.

**Outside of Hydrology:**

Coscia, I., Greenhalgh, S., Linde, N., Doetsch, J. A., Marescot, L., Günther, T., Vogt, T. & Green, A. (2011): 3D crosshole ERT for aquifer characterization and monitoring of infiltrating river water. Geophysics 76(2), G49-G59, doi:10.1190/1.3553003.

There are also a couple of papers in HESS on groundwater/river water interaction without ERI being involved.

---

## Author Comment (AC3) · 19 Jun 2018

**General Comments:**

I have prepared a concise review which you can find in attachment (annotated PDF). Overall, it is an interesting paper and topic but it lacks a proper presentation of the data (ERI) and methodology (inversion, model quality, ground-truth validation) in order to be published. I would therefore recommend moderate to major revisions.

**Author Response:** Thank you for the thorough review and constructive comments. Numerous details about the research methodology (inversion criteria, handling of underwater survey sections, model quality, survey setup) will be added to the revised manuscript as suggested. Additionally, the time-lapse profiles will include the topography used in the background ERI profiles. The time-lapse resistivity profiles will be presented using ratios of relative change to improve interpretability. Details about the laboratory testing of soil samples will be added to validate the geophysical interpretations. A detailed discussion will be added to better explain the geophysical interpretations as they relate to hydrologic processes.

Please also note the supplement to this comment:

**Specific Comments:**

https://www.hydrol-earth-syst-sci-discuss.net/hess-2018-133/hess-2018-133-RC3-supplement.pdf